# Abundance of Regulatory T Cell (Treg) as a Predictive Biomarker for Neoadjuvant Chemotherapy in Triple-Negative Breast Cancer

**DOI:** 10.3390/cancers12103038

**Published:** 2020-10-19

**Authors:** Masanori Oshi, Mariko Asaoka, Yoshihisa Tokumaru, Fernando A. Angarita, Li Yan, Ryusei Matsuyama, Emese Zsiros, Takashi Ishikawa, Itaru Endo, Kazuaki Takabe

**Affiliations:** 1Department of Surgical Oncology, Roswell Park Comprehensive Cancer Center, Buffalo, NY 14263, USA; masa1101oshi@gmail.com (M.O.); 590mariko@gmail.com (M.A.); Yoshihisa.Tokumaru@roswellpark.org (Y.T.); Fernando.AngaritaCelis@RoswellPark.org (F.A.A.); 2Department of Gastroenterological Surgery, Yokohama City University Graduate School of Medicine, Yokohama 236-0004, Japan; ryusei@yokohama-cu.ac.jp (R.M.); endoit@med.yokohama-cu.ac.jp (I.E.); 3Department of Breast Surgery and Oncology, Tokyo Medical University, Tokyo 160-8402, Japan; tishik55@gmail.com; 4Department of Surgical Oncology, Graduate School of Medicine, Gifu University, 1-1 Yanagido, Gifu 501-1194, Japan; 5Department of Biostatistics & Bioinformatics, Roswell Park Comprehensive Cancer Center, Buffalo, NY 14263, USA; li.yan@roswellpark.org; 6Center for Immunotherapy, Roswell Park Comprehensive Cancer Center, Buffalo, NY 14263, USA; Emese.Zsiros@RoswellPark.org; 7Department of Gynecologic Oncology, Roswell Park Comprehensive Cancer Center, Buffalo, NY 14263, USA; 8Department of Surgery, Jacobs School of Medicine and Biomedical Sciences, State University of New York, Buffalo, NY 14263, USA; 9Division of Digestive and General Surgery, Niigata University Graduate School of Medical and Dental Sciences, Niigata 951-8520, Japan; 10Department of Breast Surgery, Fukushima Medical University School of Medicine, Fukushima 960-1295, Japan

**Keywords:** biomarker, cytolytic activity, immune cell, regulatory T cells, tumor immune microenvironment, tumor-infiltrating lymphocyte, triple negative breast cancer, survival analysis, xCell

## Abstract

**Simple Summary:**

Regulatory CD4^+^ T cell (Treg) is one of the suppressive immune cells, but data on its clinical relevance in large human breast cancer cohort is limited. Abundance of Tregs in 5177 breast cancer patient samples from five independent cohorts was analyzed by the xCell algorithm using tumor transcriptomics. Treg abundance was lower in metastatic tumors when compared to matched primary tumors. Treg was associated with a high mutation rate of *TP53* genes and copy number mutations as well as with increased tumor infiltration of M2 macrophages and decreased infiltration of T helper type 1 cells. Interestingly, low Treg abundance was significantly associated with pathological complete response (pCR) after neoadjuvant chemotherapy (NAC) in TNBC, but not in ER-positive/Her2-negative subtype. Abundance of Treg was also associated with high expression of multiple immune checkpoint molecules. In conclusion, Treg abundance may have a potential as a predictive biomarker of pCR after NAC in TNBC.

**Abstract:**

Regulatory CD4^+^ T cell (Treg), a subset of tumor-infiltrating lymphocytes (TILs), are known to suppress anticancer immunity but its clinical relevance in human breast cancer remains unclear. In this study, we estimated the relative abundance of Tregs in breast cancer of multiple patient cohorts by using the xCell algorithm on bulk tumor gene expression data. In total, 5177 breast cancer patients from five independent cohorts (TCGA-BRCA, GSE96058, GSE25066, GSE20194, and GSE110590) were analyzed. Treg abundance was not associated with cancer aggressiveness, patient survival, or immune activity markers, but it was lower in metastatic tumors when compared to matched primary tumors. Treg was associated with a high mutation rate of *TP53* genes and copy number mutations as well as with increased tumor infiltration of M2 macrophages and decreased infiltration of T helper type 1 (Th1) cells. Pathological complete response (pCR) after neoadjuvant chemotherapy (NAC) was significantly associated with low Treg abundance in triple negative breast cancer (TNBC) but not in ER-positive/Her2-negative subtype. High Treg abundance was significantly associated with high tumor expression of multiple immune checkpoint inhibitor genes. In conclusion, Treg abundance may have potential as a predictive biomarker of pCR after NAC in TNBC.

## 1. Introduction

Tumor-infiltrating lymphocytes (TILs) have gained growing attention because they indicate a better prognosis in several types of cancer [1,2,3]. Breast cancer typically has less immune cell infiltration compared to other cancers. Triple negative breast cancer (TNBC) is the most aggressive subtype because of its worst outcome [4]; however, TNBC is also the most immunogenic subtype that attracts TILs due to genomic instability and higher mutation rate [5,6]. Tumoral abundance of TILs in breast cancer predicts response to neoadjuvant chemotherapy (NAC) and correlates with better prognosis in TNBC [7,8].

NAC for locally advanced breast cancer not only facilitates breast conserving surgery, but it also tests the tumor’s sensitivity to anti-cancer treatment, thereby guiding adjuvant treatment [9]. However, there are also reports that contradict the TILs’ effect on the response to NAC or prognosis [10]. One of the reasons for this discrepancy is that TILs are a heterogeneous cell population containing both tolerogenic regulatory CD4^+^FoxP3+ T cells (Tregs) and tumor recognizing CD8^+^ effector T cells. Currently, TILs are identified by pathology in clinical practice that does not distinguish between these different subsets of T cells.

Regulatory CD4^+^ T cells (Tregs) are a type of TILs that has gained considerable attention [11,12,13]. Tregs play a critical role in immune tolerance and they suppress antitumor immune responses [14,15]. Tumor-infiltrating Tregs suppress the proliferation of effector T lymphocytes, which prohibits an adequate anti-cancerous immune response and promotes tumor growth [16]. However, the prognostic significance of Tregs in breast cancer remains ambiguous, and data from large patient cohort studies are awaited to fully understand their clinical relevance. Tregs are mostly identified by immunohistochemistry (IHC) of Forkhead box protein 3 (*Foxp3*) protein. However, because *Foxp3+* T lymphocytes are heterogeneous cells, *Foxp3* IHC staining alone is insufficient to identify these cells [17,18,19]. Although IHC is the gold standard to identify and quantify immune cells, it is examiner-dependent and therefore results may vary and be inaccurate. Our group and others have reported the clinical relevance of the tumor immune microenvironment (TIME) using computational algorithms from bulk tumor transcriptome of the patient cohorts [20,21,22,23,24]. Utilizing xCell algorithm on transcriptomic data, we found that adipocytes in TIME of breast cancer are associated with metastasis and inflammation-related pathways particularly in ER-positive/human epidermal growth factor receptor 2 (HER2)-negative breast cancer [25]. We also found that, in breast cancer, a biologically aggressive phenotype and anti-cancerous immunity is associated with high mutation rate [22]. An advantage of this approach is that it allows the analyses of actual patient cohorts that are linked with transcriptomic data but were collected for completely unrelated motives. In this study, we hypothesized that transcriptionally defined low levels of tumor infiltration of Tregs is associated with pathological complete response (pCR) after NAC and can serve as a predicative biomarker in breast cancer patients. In this study, we analyzed 5177 breast cancer patients from five completely independent cohorts—TCGA-BRCA (*n* = 1065) [26], GSE96058 (*n* = 3273) [27], GSE25066 (*n* = 508) [28] GSE20194 (*n* = 248) [29], and GSE110590 (*n* = 83)—to study the association of Treg abundance, pathological characteristics, and clinical outcomes of patients with various breast cancer types.

## 2. Results

### 2.1. Abundance of Regulatory T Cell (Treg) in Tumors Is Not Associated with Cancer Aggressiveness

We utilized the xCell algorithm [30] to estimate the relative abundance of Tregs from transcriptome of a bulk tumor. The xCell algorithm defined Treg by expression profile of 39 genes (Appendix A). First, we were interested to find out whether stage, grade, and disease-free and disease specific survival are associated with the levels Treg infiltration in patients with different types of breast cancer. In the breast cancer cohort of The Cancer Genome Atlas (TCGA-BRCA; *n* = 1098), abundance of Tregs in primary tumors was not associated with cancer subtype, American Joint Committee on Cancer (AJCC) staging, or Nottingham pathological grade (Figure 1A; *p* = 0.640, *p* = 0.172, and *p* = 0.963, respectively).

Abundance of Tregs in various metastatic breast cancer (in brain, lung, bone, lymph node, and liver) were found to be significantly less compared with primary tumors in the GSE110590 cohort (Figure 1B; *n* = 83, *p* < 0.001). The reduction of Treg abundance defined by tumor transcriptome analysis was roughly the same for all breast cancer subtypes. Treg abundance in the metastases did not vary by site of metastasis.

To examine the association of the amount of Tregs in primary tumors and disease outcome, patients were divided into high and low Tregs groups using the top tertile as a cut-off within a cohort. Distribution of Treg was shown by histograms in the TCGA and GSE25066 cohorts (Appendix A). Abundance of Tregs in primary tumors was not associated with disease outcome as measured by disease-free survival (DFS), disease-specific survival (DSS), or overall survival (OS) in the TCGA and GSE25066 cohort (*n* = 508) (Figure 1C).

### 2.2. High Abundance of Treg Was Not Associated with Tumor Mutational Load or Enrichment of Immune Response, but Associated with Increased Infiltration of M2 Macrophages and T Helper Type 2 (Th2) Cells and Decreased Infiltration of T Helper 1 (Th1) Cells

Tumor mutational burden across several tumor types correlates with TIL infiltration and favorable response to treatment with immune check point inhibitors [31,32]. Next, we were interested to find out whether Treg infiltration is associated with tumor mutational load. To test this notion, we first examined the association of Treg infiltration and several mutation-related scores in TCGA cohort as described by Thorsson et al. [33]. High Treg infiltration was significantly associated with a high score of copy number alteration (CNA), but not silent- or nonsilent-mutation score (Figure 2A, *p* = 0.023, 0.155, and 0.271, respectively). Next, we examined the association of Treg infiltration and mutation rate of *TP53* and *PIK3CAM*, which are the top two most mutated genes in breast cancer in the TCGA cohort. High Treg infiltration was associated with a high *TP53* mutation rate, but not with the *PIK3CAM* mutation rate (Figure 2B, *p* = 0.037 and 0.945, respectively).

Given that Tregs are immunosuppressive and inhibit effector T cells proliferation [34], we also expected Treg infiltration to be associated with suppressed immune response and with less infiltration of anti-cancer immune cells in tumor immune microenvironment (TIME). To this end, gene set enrichment analysis (GSEA) of the Hallmark gene sets was performed on the TCGA and GSE25066 cohorts. Treg high abundance tumors did not enrich any of the immune-related Hallmark gene sets, such as interferon (IFN)-α response, IFN-γ response, allograft rejection, and inflammatory response, in either of the two cohorts (Figure 2C).

Next, we investigated if Treg abundance in the cohorts was associated with relative infiltration of various immune cells, which were estimated using the xCell deconvolution method on the bulk tumor transcriptome. We found that high abundance of Tregs was associated with high infiltration of M2 macrophages, which are generally considered to be immunosuppressive cells, CD8^+^ T cells, and dendritic cells (DC) and with less infiltration of T helper type 1 (Th1) cells in both cohorts (Figure 2D). Infiltrations of T helper type 2 (Th2) cells, CD4^+^ memory T cells, and M1 macrophages were also significantly higher in Treg-high tumors in either TCGA or GSE25066 cohort, but not both.

### 2.3. Pathological Complete Response (pCR) to Neoadjuvant Chemotherapy (NAC) Is Associated with Less Treg Abundance in Triple Negative Breast Cancer (TNBC) but Not in ER-Positive/Her2-Negative Breast Cancer

TILs have recently been reported to not only act as a prognostic indicator but also as a predictive biomarker for pCR to NAC, particularly in patients with TNBC [35]. Therefore, it was of interest to study the association of pCR to NAC with levels of immune cells in the primary tumor. The xCell algorithm was used to estimate infiltration of pro-cancerous, as well as anti-cancerous immune cells by bulk tumor transcriptome profiles from the GSE25066 (*n* = 508) and GSE20194 (*n* = 248) cohorts. pCR was not associated with high infiltration of either CD4^+^ or CD8^+^ T cells, which constitute the majority of TILs, in either ER-positive/Her2-negative or TNBC subtypes. pCR was significantly associated with high levels of Th1 and Th2 cells, and M1 macrophages in ER-positive/HER2-negative breast cancer in the GSE25066 cohort, but not the GSE20194 cohort (Figure 3A). For the TNBC subtype, the lack of Treg infiltration alone had significant association with pCR consistently in both patient cohorts (Figure 3B, both *p* = 0.014). These findings suggest that less abundance of Treg alone demonstrated consistent and significant association with pCR to NAC in TNBC.

### 2.4. High Abundance of Treg Was Associated with Significantly Worse pCR Rate after NAC in TNBC, but Not in ER-Positive/Her2-Negative Breast Cancer

Given that pCR after NAC was significantly associated with less infiltration of Tregs in TNBC, we investigated whether high top tertile amount of Treg could be a predictive biomarker for NAC. We found that high abundance of Treg prior to the treatment was associated with significantly lower pCR rate in TNBC patients consistently in both the GSE25066 and GSE20194 cohorts (Figure 4. *p* = 0.009, and 0.034, respectively). No association was observed between Treg and pCR rate in ER-positive/HER2-negative patients in both cohorts. These results suggest that high abundance of Treg in TNBC may be a predictive biomarker of pCR after NAC.

### 2.5. Tumors with High Abundance of Treg Are Associated with High Expressions of Immune Checkpoint Molecules

Given the recent approval of immune checkpoint inhibitors (ICIs) as treatment for some breast cancer patients [36], it was of interest to examine the association of abundance of Tregs with expression of major immune checkpoint molecules. We found that high abundance of Treg had significantly elevated gene expression of programmed death ligand 1 (*PD-L1*) and cytotoxic T-lymphocyte-associated protein 4 (*CTLA4*) in ER-positive/HER2-negative tumors, but it had significantly elevated levels of *PD-L1*, *PD-L2*, and *CTLA4* in TNBC in the TCGA cohort (Figure 5). In the GSE96058 cohort, tumors of either subtype with high abundance of Tregs had significantly elevated expression of all the immune checkpoint genes examined. These findings suggest that abundance of Tregs is associated with expression of immune checkpoint molecules.

## 3. Discussion

In this study, we investigated the clinical relevance of Treg abundance in breast cancer by estimating Treg levels using the xCell algorithm on tumor transcriptomes. There was no association of Treg abundance with cancer aggressiveness, patient survival, or immune-related pathways within tumors. Abundance of Treg in metastatic tumor was lower than that in primary tumors. High abundance of Tregs was significantly associated with high mutation rate of *TP53* gene and copy number mutation score. Treg abundance was significantly associated with increased tumor infiltration of M2 macrophages, CD8^+^ T cells, and DC and decreased infiltration of Th1 cells. Interestingly, pCR after NAC was significantly associated with low abundance of Tregs in TNBC, but not in ER-positive/HER2-negative breast cancer. No other TIL subset besides Treg was found to be associated with pCR after NAC. High abundance of Tregs significantly associated with low pCR rate consistently in two independent patient cohorts. In addition, high Treg abundance was significantly associated with high expression of multiple immune checkpoint molecules in the bulk tumors.

TILs constitute the tumor immune surveillance system. Although breast cancer is immunologically less active compared to other solid tumors, there are data suggesting TILs can predicate response to therapy and survival in breast cancer [35,37]. Recently it has been shown that not only does the quantity of TILs determine the clinical outcome, but also the types of lymphocytes [38,39,40]. For instance, the role tumoral CD4^+^ T cells, a major component of TILs, is multifaceted. CD4^+^ T cells that differentiated into Th1 cells have been shown to sustain the cancer cytotoxic functions of CD8^+^ T cells. On the other hand, Treg subset of CD4^+^ T cells have been shown to inhibit cytotoxic functions of CD8^+^T cells, support B cell growth, and promote cancer progression [41].

Due to the ability to inhibit anti-cancerous immunity, a high abundance of tumor-infiltrating Tregs were associated with worse prognosis [42,43,44,45,46]. Wang et al. have shown that high infiltration of Treg correlated with worse DFS, suggesting that Treg abundance may be a prognostic biomarker for breast cancer [18]. However, others have challenged this notion by either showing a lack of correlation of Treg abundance with survival [47] or by showing that abundance of Treg was associated with better survival in some cancers [48], including TNBC [49,50,51]. For instance, Treg identified by IHC was associated with the efficacy of NAC in breast cancer patients [52]. Furthermore, our group and others have found that abundance of Treg in metastatic tumor was lower than that in primary tumors [53,54], which appears paradoxical considering metastatic tumors are far more aggressive than the primary cancer. One of the reasons for these results may be due to the method of identifying Tregs. The most common method to identify Tregs is to stain Forkhead box protein 3 (*Foxp3*) by IHC [55,56]. The issue is that *Foxp3*-positive T cells are heterogeneous population of cells, and *Foxp3* alone is not sufficient to capture Tregs. For instance, when Tregs were identified by *Foxp3* alone, there was no association of Tregs and survival outcome of colorectal cancer. On the other hand, a positive association with patient outcome was found when *Foxp3* and B lymphocyte-induced maturation protein-1 (Blimp-1) were both used to identify Tregs [17,57]. Another example is that Tregs were associated with good outcome in B-cell lymphoma when they were identified by *Foxp3* alone, but Tregs were associated with poor patient outcome when they were identified by both *CTLA-4* and Foxp3 [57]. Zhang et al. reported that staining with Foxp3 together with *CD25* is better than *Foxp3* alone to evaluate Tregs [58]. These examples eloquently demonstrate that multiple markers are necessary to identify biologically active Tregs in human cancer [17,18,19]. Our group has previously reported the clinical relevance of specific cells in the TIME by using multiple gene expression profiles with algorithms [59,60,61,62]. This approach was feasible and efficient in analyzing large patient cohorts because we utilized transcriptome of publicly available large cohorts that are generated for different research objectives. The gold standard to analyze immune cells is IHC or flow cytometry. Nonetheless, IHC is limited by its ability to only quantify cells and user dependent variation of results. Flow cytometry requires fresh sample, which is usually very challenging to obtain. Moreover, both methods are highly costly and labor intensive compared to our in silico approach. We believe that current study can be one of the breakthroughs of the field since we estimated the relative abundance of Tregs in breast cancer by using the xCell algorithm to analyze gene expression data of bulk tumors of multiple large patient cohorts. In addition to Tregs, the xCell algorithm can be used to infer the relative abundance of 63 other types of immune and stromal cells using the known gene signatures of each cell types [30]. We further found that, although the amount of Tregs was not associated with survival outcome, pCR after NAC was significantly associated with Tregs and not with any other immune cells in TNBC. Furthermore, the Treg abundance showed a potential as a predictive biomarker for NAC response in TNBC. No other immune cells showed similar results in the two independent cohorts. One of the reasons may be due to differences in NAC treatment regimens between cohorts. Taxane and anthracycline regimen was used in the GSE25066 and paclitaxel, 5-fluorouracil, cyclophosphamide and doxorubicin regimen was used in the GSE20194 cohort. Further verification is required to conclude the association between immune cells and treatment response.

Because each breast cancer subtypes is biologically different, the impact of TILs on clinical outcome should be evaluated separately in each subtype [63]. TNBC is known to be both the most aggressive breast cancer subtype as well as the most “immunomodulatory subtype” because it has elevated immune cell infiltration. Although ICIs are approved for TNBC, it is effective to very limited population and the patient selection remains a major challenge. Given that our results show that the majority of immune checkpoint molecules were associated with abundance of Tregs not only in TNBC but also in ER-positive/HER2-negative breast cancer, we cannot help but speculate that patients whose tumors show Treg abundance could be the population who would respond to ICIs.

This study has some limitations. The biggest limitation is that our results are based on analyses of tumor gene expression alone without any direct quantification of tumoral Tregs using gold standard such as flow cytometry and/or immunohistochemistry, which are analyses in protein level. Our approach allowed us to analyze very large sample number cohorts; however, we were unable to validate our results at protein level because we do not have physical access to the samples of these cohorts. To this end, further studies using protein levels are needed to investigate the clinical applicability of our findings in the future. This is a retrospective study that used publicly available cohorts of multiple previous studies. The cohorts we examined vary between each other by clinical characteristics. Additionally, data regarding co-morbidities and therapeutic information were lacking. Furthermore, our analyses are limited to exact spatial location of where the sample was taken by the original authors. For breast cancer, TILs can be divided into intratumoral and stromal subtypes according to their localization in the tumor tissue. The International TILs Working Group recommended the use of stromal TILs to assess the significance of TILs in breast cancer [64], whereas our samples are more likely to be taken from inside the tumor. Recently, some claim that effector T cell/Treg ratio is important for survival and drug response; however, we were unable to conduct this analysis due to technical difficulty. Finally, this study did not assess biological mechanisms that underlie its clinical findings.

## 4. Materials and Methods

### 4.1. Data of Breast Cancer Cohorts

Data for tumor gene expression and corresponding clinical information for the TCGA-BRCA (*n* = 1065) [26] cohort were obtained from the cBio Cancer Genomic portal [65]. Mutation data (*TP53* and *PIK3CAM*) of the TCGA cohort were also obtained from the cBio Cancer Genomic portal, as we previously reported [66]. TCGA was chosen as the main cohort for our analyses because it is one of the largest breast cancer cohorts with robust tumor transcriptome and clinical data. However, one of the weaknesses of the TCGA is that the treatment data are limited, and it is assumed that all the patient underwent “standard of care”. The Gene Expression Omnibus (GEO) repository was used to access the tumor gene expression and clinical data from the Brueffer et al. (GSE96058; *n* = 3273) [27], Symmans et al. (GSE25066; *n* = 508) [28], Shi et al. (GSE20194; *n* = 248) [29], and Siegel et al. (GSE110590; *n* = 83) [67] cohorts. GSE25066 was chosen as the validating cohort of TCGA cohort; however, it needs to be noted that the sample size of GSE25066 is about a half of TCGA. GSE25066 and GSE20194 cohorts were chosen because these two cohorts were among the largest cohorts with patient NAC response data that we identified. GSE110590 cohort was used to investigate the association of Treg infiltration with metastases because it has transcriptomic data for metastatic breast cancer, of which was not available in any other cohorts. Finally, we used the GSE110590 cohort to investigate the association of Treg infiltration with expression of multiple immune checkpoint molecules because in-depth coverage of transcriptome was available in this cohort. This information is shown in Appendix A. We believe that having multiple completely independently reported large cohorts support our hypothesis, making our argument even stronger. The average of the signals from the multiple probes was used to analyze genes with multiple hybridization probes in microarray-based gene expression data. Gene expression data were transformed for log_2_ in all analyses. The top tertile was defined as the high Treg infiltration within cohorts.

### 4.2. Tumor Immune Cell Scoring Using xCell

The tumor Tregs score was determined with the xCell algorithm [30], using bulk tumor gene expression profiles. The xCell algorithm was used for tumor composition analysis of infiltrated immune cells, as we previously reported [68,69,70]. Genes used to calculate each immune cell are shown in Appendix A.

### 4.3. Gene Set Enrichment Analysis

Gene set enrichment analysis were performed with Gene Set Enrichment Analyses (GSEA) software (Java version 4.0) [71,72] with MSigDB Hallmark gene sets [73]. Statistical significance was set with a false discovery rate (FDR) of 0.25, as recommended by the GSEA software and as we previously reported [66,74,75,76,77,78,79,80].

### 4.4. Other

R software (version 4.0.1, R Project for Statistical Computing) and Excel (version 16 for Windows; Microsoft, Redmond, WA, USA) were used for data analyses. Within patient cohorts, those in the top tertile for tumor Tregs score were considered to be in the high Tregs group, while the other were grouped into the low Tregs group. Gene expression data were transformed for log_2_ in all analyses. Kruskal-Wallis test was used to compare multiple groups. Mann-Whitney U test and Fisher’s exact tests were used to compare two groups. Survival analyses were performed using the Kaplan–Meier method with log-rank test.

## 5. Conclusions

pCR after NAC was significantly associated with low Treg abundance in TNBC, but not in ER-positive/HER2-negative subtype. Thus, we conclude that Treg abundance may have a potential as a predictive biomarker of pCR after NAC in TNBC.

## Figures and Tables

**Figure 1 cancers-12-03038-f001:**
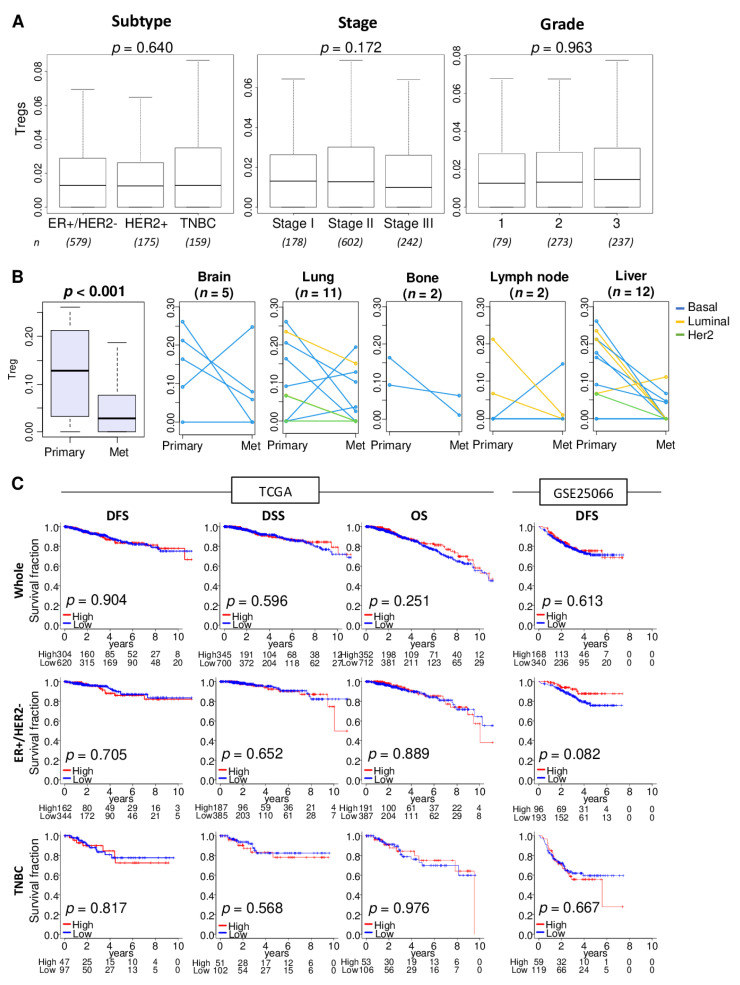
Association of Tregs abundance with clinical characteristics, metastasis, and survival outcomes. (**A**) Boxplots of Tregs level distribution by subtype, American Joint Committee on Cancer stages, and Nottingham pathological grade in the TCGA cohort. Kruskal-Wallis test was used for the analysis. Group sizes are shown underneath the plots. (**B**) Boxplots of the Tregs levels in the primary breast cancer and different metastatic tumors. Matched pair comparison of Treg levels between the primary and each metastatic tumor; brain, lung, bone, lymph node, and liver in the GSE110590 cohort. Mann-Whitney U test was used to the analysis. Blue line; Basal type, yellow line; Luminal type, and green line; Her2 type. (**C**) Disease-free survival (DFS), disease-specific survival (DSS), and overall survival (OS) stratified by Tregs levels (low (blue line) and high (red line)) within whole tumors, estrogen receptor (ER)-positive/human epidermal growth factor receptor 2(HER2)-negative, and triple negative breast cancer (TNBC) in the TCGA and GSE25066 cohorts. The top tertile percent was used as a cut-off between high and low Tregs groups within each cohort. Log-rank test with Kaplan-Meier survival curves was used for the survival analysis.

**Figure 2 cancers-12-03038-f002:**
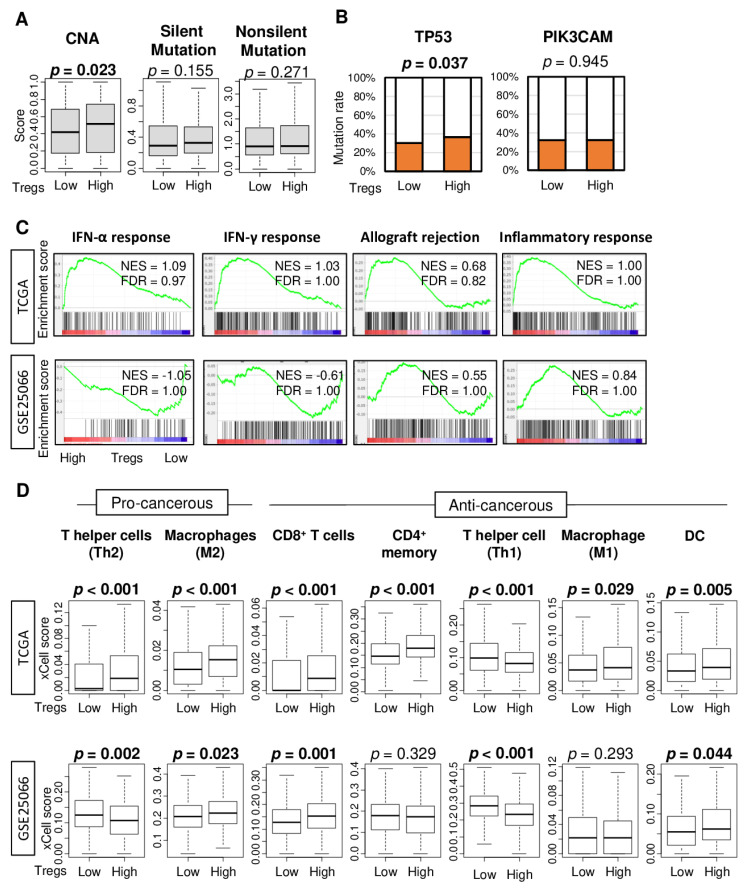
Mutation, immune-related gene sets, and tumor infiltrating immune cell compositions by low and high abundance of Treg in the TCGA and GSE25066 cohorts. (**A**) Boxplots of comparison between low and high Treg groups with mutation-related score; copy number alteration (CNA), silent- and nonsilent-mutation score in TCGA cohort. Mann-Whitney U test was used for the analysis. (**B**) Bar plots comparing low and high Treg groups with *TP53* and *PIK3CAM* genes mutation rates in the TCGA cohort. Fisher’s exact test was used for the analysis. (**C**) Enrichment plots for Hallmark gene sets for which highly enriched in the Treg high tumors compared to low tumors in the TCGA and GSE25066 cohorts, along with normalized enrichment score (NES) and false discovery rate (FDR). The statistical significance of GSEA was determined by FDR < 0.25. (**D**) Boxplots comparing low and high Treg groups with infiltrating immune cell compositions using xCell algorithm in breast cancer. Pro-cancerous immune cells included T helper type 2 cells (Th2), and M2 macrophages. Anti-cancerous immune cells included CD8^+^ T cells, CD4^+^ memory T cells, T helper type 1 cells (Th1), and M1 macrophages. The top tertile percent was used as a cut-off between low and high Treg group. Mann-Whitney U test was used for the analysis.

**Figure 3 cancers-12-03038-f003:**
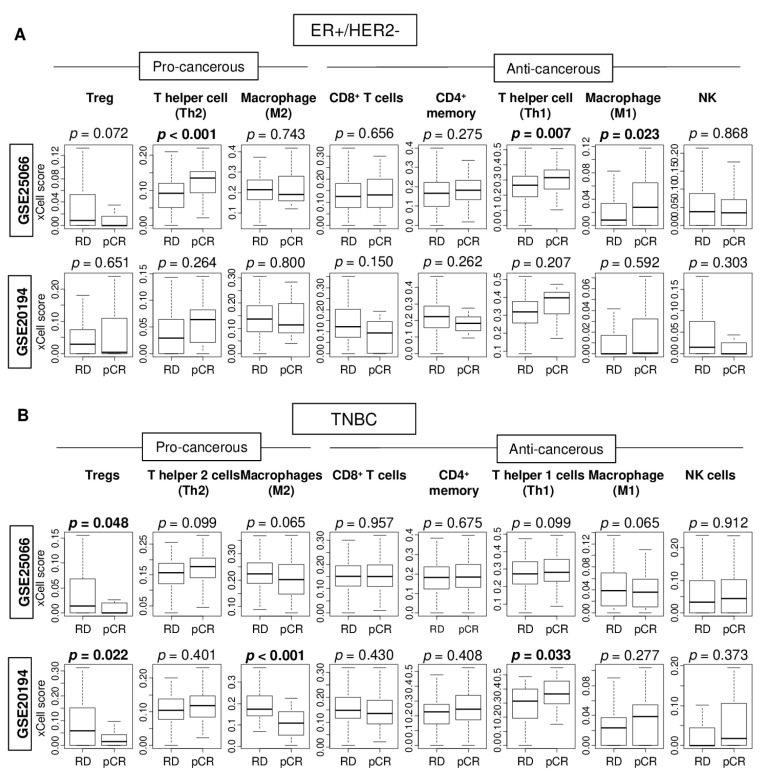
Association of pathological complete response (pCR) with fraction of multiple tumor infiltrating immune cells in the GSE25066 and GSE20194 cohorts. These cohorts were chosen because of their response to neoadjuvant chemotherapy data. Boxplots of levels of immune cells with residual disease (RD) and pCR after NAC in (**A**) ER-positive/HER2-negative tumors and (**B**) TNBCs in both patient cohorts. Mann–Whitney U test was used for the analysis. Pro-cancerous immune cells included Tregs, T helper type 2 cells (Th2), and M2 macrophages. Anti-cancerous immune cells included CD8 T cells, CD4 memory T cells, T helper type 1 cells (Th1), M1 macrophages, and Natural Killer T cells (NK).

**Figure 4 cancers-12-03038-f004:**
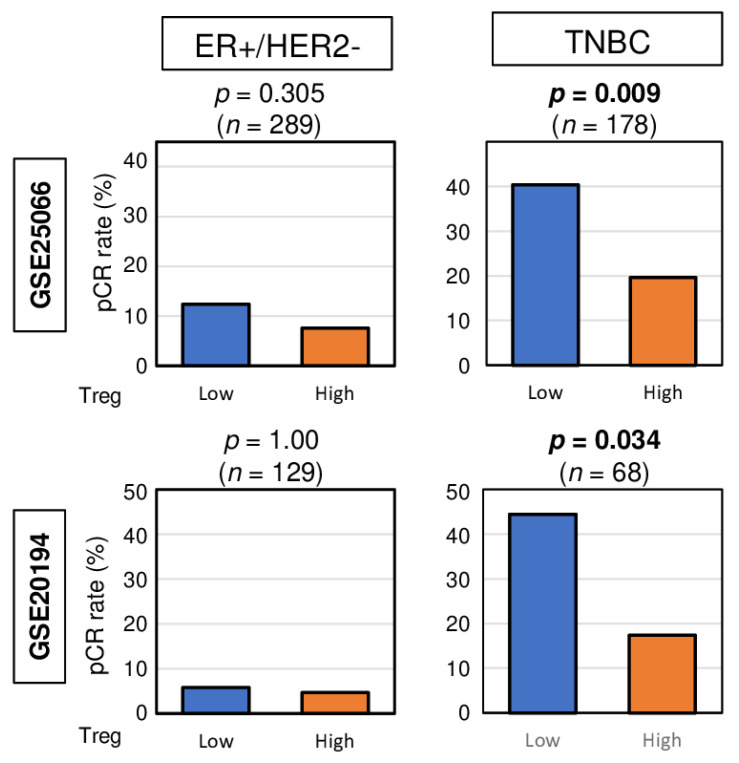
Bar plots depicting the pCR rates between low and high abundance of Treg groups among patients with ER-positive/HER2-negative tumors and TNBCs in the GSE25066 (*n* = 508) and GSE20194 (*n* = 248) cohorts. These cohorts were chosen because of their response to neoadjuvant chemotherapy data. Fisher’s exact test was used for the analysis.

**Figure 5 cancers-12-03038-f005:**
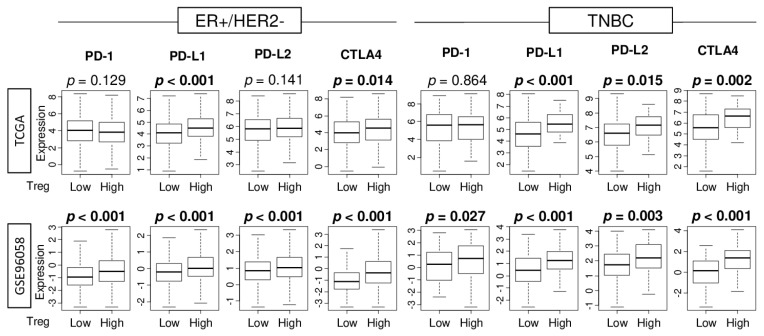
Association of Treg infiltration with expression of immune checkpoint molecules. Boxplots comparing low and high infiltration of Tregs in gene expression of immune checkpoint molecules in ER-positive/HER2-negative and TNBC patients in the TCGA and GSE96058 cohorts which have the PD-L1 (*CD274*) expression data. Mann-Whitney U test was used for the analysis.

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
