# Peer review of "Abundance of Regulatory T Cell (Treg) as a Predictive Biomarker for Neoadjuvant Chemotherapy in Triple-Negative Breast Cancer"

_cancers, 2020, doi:10.3390/cancers12103038_

Round 1

Reviewer 1 Report

This study examines the roles of regulatory T cells (Treg) and other immune cells in five breast cancer cohorts by using in silico methods and finds that high Treg abundance may serve as a biomarker of pathologic complete response (pCR) after neoadjuvant chemotherapy (NAC) in triple negative breast cancer. Although the results sound significant, the following questions should be considered:

  1. Statistical analysis. The authors use ANOVA and Fisher's exact test to examine the difference between groups (compare the mean of each category). However, some data are continuous variables, such as gene expression levels in Fig 5, xCell score in Figs 2D and 3, mutation score in Fig 2A, Treg % in Fig 1A and 1B, which should be examined by other tests, such as Mann-Whitney U test?
  2. Different cohorts are used for different examinations (clinical parameters, NAC response, TIL cell types,…) without explanation (lacking some of these information?). For an example, in Fig 3, no NAC data in TCGA cohort? It would be better to have a summary table showing the availability of these information for each cohort.
  3. In Fig 2D, Th2 score is higher in TCGA but is lower in GSE25066 cohort. This opposite result is not as the authors mentioned in lines 142-143 “We found that high abundance of Tregs was associated with high infiltration of other immunosuppressive cells, such as M2 macrophages and T helper type 2 (Th2) cells”
  4. Lines 222-223 (Fig 3A), Th1, M1 may be associated with pCR in certain cohort, such as GSE25066, although the statistical significance may need to be verified. As the authors discussed, different cohorts may have different clinical features that lead to different results. Thus, additional study is required for obtaining a solid conclusion.
  5. The result section in lines 95-99 should be denoted by Fig 1B. Lines 308-310, citation #71 for GSEA method; PNAS 2005, 102 (43) 15278-15279.

Reviewer 2 Report

Major comments

  1. The conclusion of this study is solely based on the bulk RNAseq data analysis of cancer patient tissue. The finding is at RNA transcriptional levels, the authors do not perform any validation experiments at protein level. To convince the audience, the authors need provide some validation data in the manuscript.
  1. Authors claim they performed the analysis on 5 cohorts, but each figure shows selected cohorts’ data. Since those cohort’s raw data are publicly available, the authors need to provide the analyzed data for all of 5 cohorts or provide a rationale why they chose certain cohorts for each figure.

Minor comments:

  1. The authors only show the gene list for Tregs in current version of the manuscript. Although authors cite the reference for identification of other immune cell subsets in their analysis, it will be more convenient for the audience if the authors could also provide the gene lists for those cellular subsets in the supplementary materials.

Round 2

Reviewer 1 Report

The authors have answered all my questions.

Reviewer 2 Report

The authors have addressed my concerns, just forgot to add gene list for DC in table S2.
